# Mental Health and Quality of Life among Patients with Cancer during the SARS-CoV-2 Pandemic: Results from the Longitudinal ONCOVID Survey Study

**DOI:** 10.3390/cancers14041093

**Published:** 2022-02-21

**Authors:** Emiel A. De Jaeghere, Heini Kanervo, Roos Colman, Wim Schrauwen, Paulien West, Nele Vandemaele, Aglaja De Pauw, Celine Jacobs, Ingeborg Hilderson, Michael Saerens, Nora Sundahl, Katrien Vandecasteele, Eline Naert, Lore Lapeire, Vibeke Kruse, Sylvie Rottey, Gilbert Lemmens, Hannelore G. Denys

**Affiliations:** 1Medical Oncology Department, Ghent University Hospital, 9000 Ghent, Belgium; heini.kanervo@uzgent.be (H.K.); aglaja.depauw@ugent.be (A.D.P.); celine.jacobs@uzgent.be (C.J.); ingeborg.hilderson@uzgent.be (I.H.); michael.saerens@ugent.be (M.S.); eline.naert@ugent.be (E.N.); lore.lapeire@uzgent.be (L.L.); sylvie.rottey@ugent.be (S.R.); hannelore.denys@ugent.be (H.G.D.); 2Cancer Research Institute Ghent (CRIG), 9000 Ghent, Belgium; nora.sundahl@ugent.be (N.S.); katrien.vandecasteele@ugent.be (K.V.); vibeke.kruse@ugent.be (V.K.); 3Biostatistics Unit, Faculty of Medicine and Life Sciences, Ghent University, 9000 Ghent, Belgium; roos.colman@ugent.be; 4Medical Psychology Department, Ghent University Hospital, 9000 Ghent, Belgium; wim.schrauwen@uzgent.be; 5Faculty of Medicine and Life Sciences, Ghent University, 9000 Ghent, Belgium; paulien.west@ugent.be (P.W.); nele.vandemaele@ugent.be (N.V.); 6Radiotherapy Department, Ghent University Hospital, 9000 Ghent, Belgium; 7Medical Oncology Department, General Hospital Sint-Niklaas, 9100 Sint-Niklaas, Belgium; 8Psychiatry Department, Ghent University Hospital, 9000 Ghent, Belgium; gilbert.lemmens@ugent.be

**Keywords:** mental health, quality of life, cancer, SARS-CoV-2, coronavirus 2019, pandemic

## Abstract

**Simple Summary:**

Although the SARS-CoV-2 pandemic is likely to have created or aggravated mental health symptoms in cancer patients, high-quality longitudinal data on this topic are scarce. The aim of our prospective survey study was to assess cancer patient-reported mental health and quality of life (QOL) at four time points during the first two waves of the pandemic. We found that an important proportion of the 355 participants reported symptoms of COVID-19 peritraumatic distress (34.2% to 39.6%), depression (27.6% to 33.5%), anxiety (24.9% to 32.7%), and stress (11.4% to 15.7%) at any time point during the study period. However, we did not find clinically meaningful mental health and QOL changes during the study period. Additionally, we found no factors associated with better or worse mental health or QOL. In conclusion, the cancer patients who participated in this study showed considerable resilience against mental health and QOL deterioration during the pandemic.

**Abstract:**

Purpose: This longitudinal survey study aimed to investigate the self-reported outcome measures of COVID-19 peritraumatic distress, depression, anxiety, stress, quality of life (QOL), and their associated factors in a cohort of cancer patients treated at a tertiary care hospital during the SARS-CoV-2 pandemic. Methods: Surveys were administered at four time points between 1 April 2020 and 18 September 2020. The surveys included the CPDI, DASS-21, and WHOQOL-BREF questionnaires. Results: Survey response rates were high (61.0% to 79.1%). Among the 355 participants, 71.3% were female, and the median age was 62.2 years (IQR, 53.9 to 69.1). The majority (78.6%) were treated with palliative intention. An important proportion of the participants reported symptoms of COVID-19 peritraumatic distress (34.2% to 39.6%), depression (27.6% to 33.5%), anxiety (24.9% to 32.7%), and stress (11.4% to 15.7%) at any time point during the study period. We did not find clinically meaningful mental health and QOL differences during the study period, with remarkably little change in between the pandemic’s first and second wave. We found no consistent correlates of mental health or QOL scores, including cancer type, therapy intention, and sociodemographic information. Conclusion: This cohort of cancer patients showed considerable resilience against mental health and QOL deterioration during the SARS-CoV-2 pandemic.

## 1. Introduction

In late 2019, the first pneumonia cases of unknown origin were identified in Wuhan (Hubei, China) [1]. The pathogen was identified as a novel enveloped ribonucleic acid human β-coronavirus that was named severe acute respiratory syndrome coronavirus 2 (SARS-CoV-2) [1]. The clinical spectrum of its associated disease, coronavirus disease 2019 (COVID-19), ranges from self-limiting upper respiratory tract illness to life-threatening pneumonia, multiorgan failure, and death [2]. Given the rapid and global spread of SARS-CoV-2, the World Health Organization (WHO) officially declared it a pandemic in March 2020 [3]. Since then, the SARS-CoV-2 pandemic and its myriad containment measures have affected all aspects of society, including cancer care across the entire disease trajectory [4,5].

The prevalence of mental health symptoms (such as depressive symptoms, anxiety, and stress) in cancer patients is known to be significant [6]; nevertheless, the SARS-CoV-2 pandemic is likely to have induced or exacerbated them in this vulnerable population. For instance, diagnostic delays [7,8], treatment modifications [9], the cancellation of in-office visits [10,11], and the reduction in provision of non-urgent care (e.g., psychosocial support and rehabilitation) might have prompted concerns about cancer recurrence, progression, and death among patients [4,12]. In addition, unwillingness to attend hospital appointments because of a perceived epidemiological threat or a reluctance to burden the overstretched healthcare system might have impaired the mental health and well-being of cancer patients [12]. Further, due to media coverage about the optimal allocation of limited healthcare resources, including ventilators, patients with active cancer might have been concerned about possible prioritization should such resources become too scarce [13]. The ESMO Resilience Task Force survey study indicated that the SARS-CoV-2 pandemic also had an impact on wellbeing, burnout, and job performance among oncology professionals [14], which, in turn, might have adversely affected the interpersonal aspects of care and cancer patient satisfaction [15]. Finally, as for the general population, many other SARS-CoV-2-related factors might have affected the mental health of cancer patients, including unemployment, financial insecurity, changes to daily routines, the practice of physical distancing (perhaps even more adhered to than in the general population), altered household dynamics (such as relationship break-ups and domestic violence), and the lack of support networks.

While there has been a surge in research examining the potential mental health consequences of living through the SARS-CoV-2 pandemic [16], to date, most studies had a cross-sectional design. Recent meta-analyses of mental health symptoms among cancer patients during the pandemic have found pooled prevalences of 37% to 43% for depressive symptoms and 38% to 53% for anxiety [17,18]. However, each stage of the pandemic has its own characteristic properties that can interfere with the interpretation of such estimates (e.g., number of infections, critical care bed capacity, media coverage, etc.). Unfortunately, data on the longitudinal patterns of mental health symptoms and quality of life (QOL) in cancer patients during the pandemic are scarce, potentially leading to bias that may exaggerate or underestimate the pandemic’s true burden.

In the present prospective cohort study, the research objectives were to longitudinally investigate trends in (i) mental health symptoms and (ii) QOL in a cohort of cancer patients during the first five months of the SARS-CoV-2 pandemic, as well as (iii) to understand the factors associated with mental health and QOL. We hypothesized that mental health and QOL would progressively worsen (to a clinically significant degree) as the pandemic continued.

## 2. Materials and Methods

### 2.1. Study Design

The ONCOVID survey study was a prospective longitudinal cohort study in cancer patients at a single tertiary care hospital in Belgium during the first and second wave of the pandemic, at four different time points. The survey period of T0 was from 1 April to 10 April 2020, T1 was from 11 May to 15 May 2020, T2 was from 22 June to 26 June 2020, and T3 was from 14 September to 18 September 2020 (Appendix A).

Surveys were completed on paper or via the internet using Research Electronic Data Capture (REDCap), a secure web-based software platform designed to support data capture for research studies [19,20]. Surveys included the COVID-19 Peritraumatic Distress Index (CPDI), the 21-item Depression, Anxiety, and Stress Scale (DASS-21), and the World Health Organization Quality of Life-BREF (WHOQOL-BREF) questionnaires. Participants also reported sociodemographic information including age, gender, highest educational level, marital status, and household composition. Treating physicians abstracted the electronic health records of eligible individuals for cancer type, therapy type, intent of therapy received, estimated life expectancy, and whether therapy was modified due to the SARS-CoV-2 pandemic.

The study was prospectively registered with the ClinicalTrials.gov registry, identifier NCT04340219. The paper is reported following the STROBE statement [21].

### 2.2. Study Sample

Eligible participants included individuals who were aged 18 years or older (with no upper age limit), had histologically confirmed cancer, and were prescribed systemic anticancer therapy at the Medical Oncology Department of Ghent University Hospital between 14 February and 31 March 2020. Key exclusion criteria were insufficient understanding of the Dutch language, severe cognitive impairment, acute psychiatric crisis, receiving adjuvant endocrine monotherapy only, and confirmed or clinically suspected COVID-19. A list of potentially eligible participants was built up from the hospital and practice medical charts and cross-checked with the list of patient contacts. 

Informed verbal and/or signed consent was obtained from each participant before inclusion. Specific follow-up or referral to mental health resources for participants who reported high levels of emotional disturbance was provided. The study was conducted according to the guidelines of the Declaration of Helsinki and approved by the local Institutional Review Board of Ghent University Hospital (BC-07505).

### 2.3. Questionnaires

The CPDI is a new 24-item self-report questionnaire designed to measure peritraumatic distress during the SARS-CoV2 pandemic [22]. The Dutch version was achieved using a process of forward–backward translation by an experienced team of health professionals working independently. The scores have the following interpretation: normal (0–28), mild-to-moderate (29–52), and severe (53–100) peritraumatic distress [22]. 

The DASS-21 is a set of three 7-item self-report subscales designed to measure the negative emotional states of depression, anxiety, and stress [23]. The scores were interpreted as follows: depressive symptoms—normal (0–4), mild, (5–6), moderate (7–10), severe (11–13), and extremely severe (14–21); anxiety—normal (0–3), mild (4–5), moderate (6–7), severe (8–9), and extremely severe (10–21); stress—normal (0–7), mild (8–9), moderate (10–12), severe (13–16), and extremely severe (17–21). These categories were based on values established in the literature [24]. 

From here on, peritraumatic distress, depressive symptom, anxiety, and stress scores will be together referred to as mental health scores. The dichotomous classification of mental health scores was defined by a score exceeding the normal limit (i.e., abnormal vs. normal scores). 

The self-report WHOQOL-BREF questionnaire measures the following four QOL domains: physical health, psychological health, social relationships, and environment [25,26,27]. The four domains encompass 24 items; all domain scores range from 0 to 100 [25,26,27]. In addition, there are two items that measure self-perceived overall QOL and general health, which are scored separately with a score range from 0 to 10 [25,26,27]. Higher values represent better QOL (scaled in a positive direction) [25,26,27]. No WHOQOL-BREF cut-offs are specified; thus, no thresholds were applied, and QOL scores were strictly defined as continuous variables in this study. 

### 2.4. Key Definitions

Age was defined as a continuous variable. Gender was defined as a categorical variable with three groups: female, male, and other (including non-binary and genderqueer). Highest educational level was defined as a categorical variable with two groups: fundamental or secondary education (preschool, primary school, and secondary education); and higher education (university and non-university formats). Marital status was defined as a categorical variable with three groups: single or never married; married or living as married; and divorced, separated, or widowed. Household composition was defined as a categorical variable with two groups: living with others and living alone. Cancer type was defined as a categorical variable with seven groups: breast; genitourinary; gynecological; melanoma; head and neck; soft tissue and bone; and other. Therapy intention was defined as a categorical variable with two groups: palliative and curative. Estimated life expectancy was defined as a categorical variable with two groups: >1 year and ≤1 year. Whether therapy changed (due to the SARS-CoV-2 pandemic) was defined as a categorical variable: yes or no. 

### 2.5. Study Endpoints

Coprimary endpoints were the proportions of participants with abnormal mental health scores at T0 (as categorical variables).

Secondary endpoints were mental health scores at T0 (as continuous variables), quality-of-life scores at T0, and the change from T0 in mental health and quality-of-life scores (at T1, T2, and T3).

*Post hoc* exploratory endpoints were the proportions of participants with abnormal mental health scores at T1, T2, and T3 (as categorical variables).

### 2.6. Statistical Analysis

We estimated the study population as 300 to 350 individuals. A response rate of 50 to 65% was targeted, which resulted in an estimated sample size of 180 patients. For the coprimary endpoints, this sample size of 180 allowed for generalizability to the study population based on a 95% Wilson score confidence interval (CI) with a half width of approximately 7.5%.

Wilson score intervals were used as CIs for the prevalences. A linear mixed model (LMM) was used for longitudinal data analysis of the mental health and QOL scores, with the survey period as a categorical predictor variable to calculate fixed effects. The model structure contained four survey periods and an unstructured variance–covariance matrix was applied to account for the dependence between the different measurements. An advantage of the LMM is that it properly accounts for missing data (at random) [28]. The results of the LMM were reported as estimated (difference in) mean scores with the corresponding 95% CI. For all scores, differences from baseline in mean score of at least half a standard deviation (SD) (of baseline data) were considered clinically significant (minimal important difference, MID) [29]. 

To assess the association of the nine participant characteristics (age, gender, educational level, marital status, household composition, cancer type, therapy intention, estimated life expectancy, and therapy change) with the scores, each characteristic was entered separately as fixed factor in the above-described LMM. Subsequently, a multivariable linear mixed regression model was used (without a variable selection method). 

We examined descriptive statistics to ensure that the data met statistical (test) assumptions. Non-response bias was assessed by comparing baseline characteristics between the participant and never-participant cohort. The nominal level of significance for all analyses was a *p* value of less than 0.05, and all hypothesis tests were two-tailed. Data analysis was performed using R (version 4.0.2) from 15 February to 4 August 2021 [30]. 

### 2.7. Role of the Funding Source

The funder (*Kom Op Tegen Kanker*, a Belgian not-for-profit organization) had no role in study design, data collection, data analysis, data interpretation, or in the writing of the report. The corresponding author (E.A.D.J.), the statistician (R.C.), and the supervisor (H.G.D.) had full access to all the data in the study. The corresponding author and supervisor shared final responsibility for the decision to submit the study for publication.

## 3. Results

### 3.1. Study Sample Characteristics

Of the 452 consecutive individuals examined for eligibility, 417 were confirmed eligible, and 390 consented to participate (Figure 1). The survey response rates were 79.1% (330 participants of 417 known eligible individuals), 75.3% (311 of 413), 70.8% (291 of 411), and 61.0% (244 of 400) for T0 to T3, respectively. The denominator (i.e., eligible individuals) varied with survey period since individuals who died after study initiation were considered ineligible from the date of death (or moribund status). Overall, 85.1% (355 of 417) of the individuals participated in at least one survey period, whereas 46.3% (185 of 400) participated in all four survey periods.

Baseline characteristics are described in Table 1. The median age of the participants was 62.2 (interquartile range [IQR], 53.9 to 69.1) years; 253 of 355 (71.3%) were female. The most common diagnoses were breast (154 of 355; 43.4%), genitourinary (56 of 355; 15.8%), and gynecological (51 of 355; 14.4%) cancer. The majority of participants were treated with palliative intention (279 of 355; 78.6%) and had an estimated life expectancy of more than one year (273 of 355; 76.9%). The never participants did not differ significantly from participants in terms of baseline characteristics.

Descriptive means and SDs as well as the MIDs of the scores at baseline are described in Appendix A. The correlation coefficients between different scores are given in Appendix A; the directions of correlations were as expected. 

### 3.2. Mental Health Scores

At T0, the following proportions of participants experienced symptoms of COVID-19 peritraumatic distress, depression, anxiety, and stress: 39.7%, (95% CI, 34.7 to 44.9%), 27.6% (95% CI, 23.1 to 32.7%), 24.9% (95% CI, 20.6 to 30.0%), and 11.4% (95% CI, 8.4 to 15.3%), respectively. The longitudinal course of the mental health score distributions during the study period is displayed in Figure 2 and Figure 3. 

At T0, a total of 10.9% (95% CI, 8.0 to 14.7%) of the participants experienced at least severe symptoms of any mental health score, whereas 1.5% (95% CI, 0.7 to 3.5%) experienced at least severe symptoms of all four mental health scores; at T1, 12.9% (95% CI, 9.7 to 17.0%) and 1.6% (95% CI, 0.7 to 3.6%), respectively; at T2, 14.4% (95% CI, 10.8 to 19.0%) and 0.7% (95% CI, 0.2 to 2.5%), respectively; and at T3, 12.0% (95% CI, 8.5 to 16.8%) and 1.2% (95% CI, 0.4 to 3.6%), respectively.
Figure 1ONCOVID study flow diagram. All the exclusion boxes in the right margin originate from the start; the numbers within these boxes should be subtracted from 390 and 417 to obtain the responders and eligible individuals at the given survey period, respectively (e.g., at T2, 291 [=390-6-20-62-11] responders out of 411 [=417-6] eligible individuals). All eligible individuals were invited to complete each survey. Questionnaires were optional; in such cases, responders may vary because of non-response.
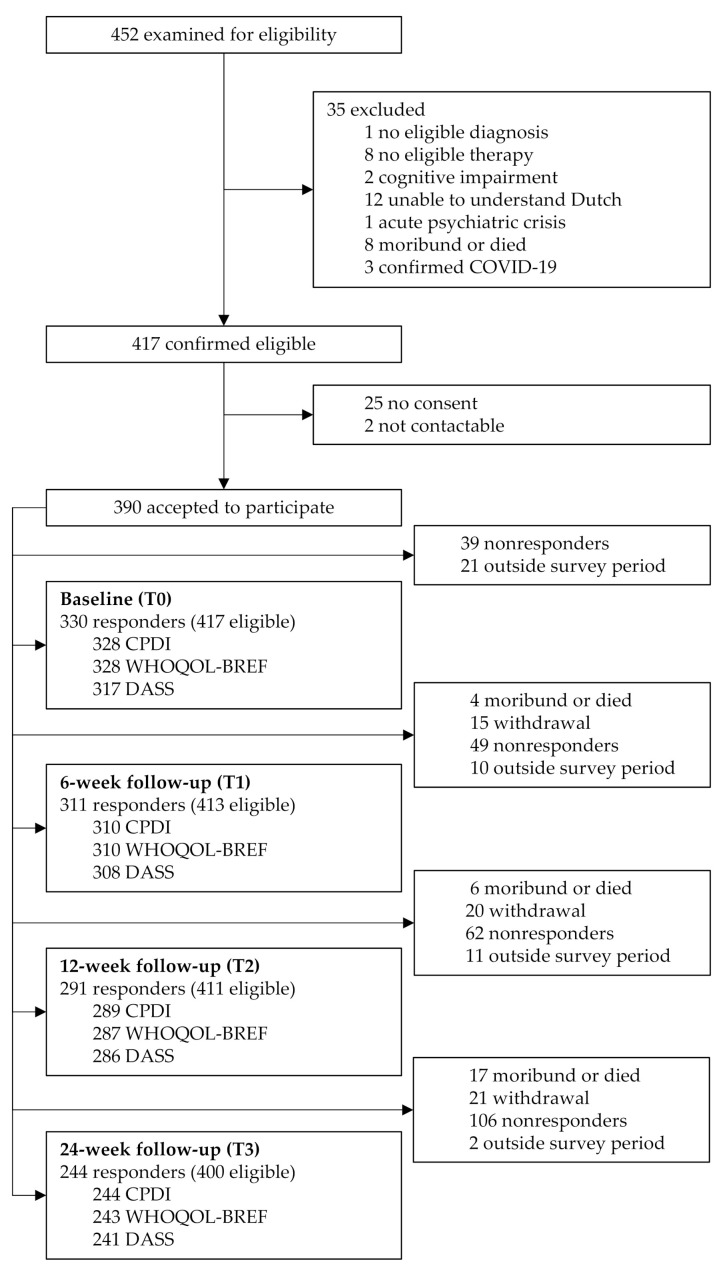

Figure 2Categorized scores for COVID-19 peritraumatic distress by survey period. Stacked bars represent the score distributions when categorized for COVID-19 peritraumatic distress severity according to the survey period. No formal hypothesis testing was performed on categorized scores. Corresponding numerical values are shown in Appendix A.
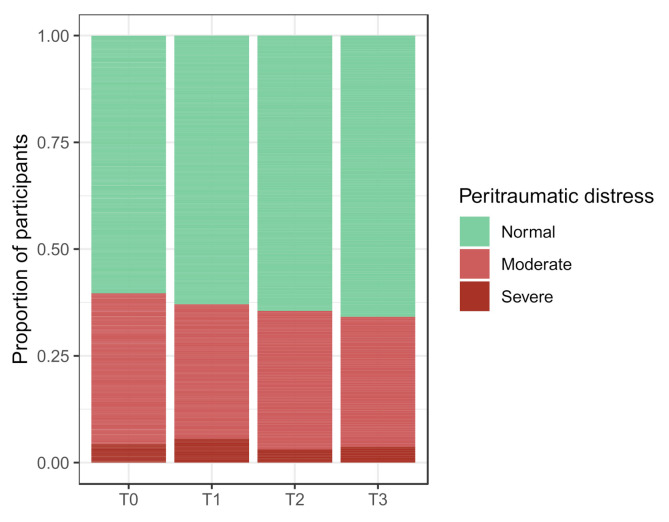

Figure 3Categorized scores for COVID-19 depression, anxiety, and stress by survey period. The same caption as that of Figure 2 applies here.
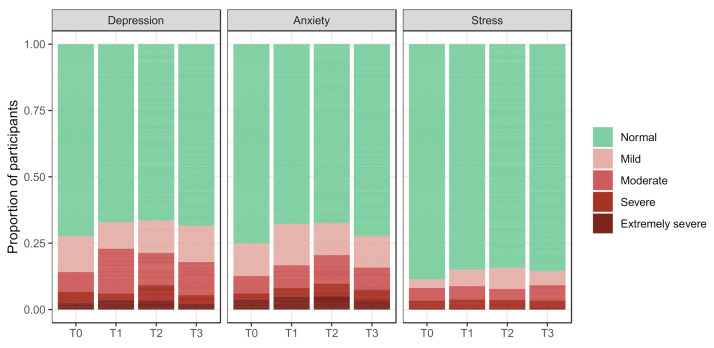

cancers-14-01093-t001_Table 1Table 1Baseline characteristics of eligible individuals according to participating status.VariableParticipants (*n* = 355)Never Participants (*n* = 62)*p* Value**Age, years**62.2 (53.9 to 69.1)61.7 (53.7 to 72.5)1.000**Gender**

1.000Female253 (71.3)44 (71.0)
Male102 (28.7)18 (29.0)
Other00
Missing00
**Educational level**

1.000Fundamental or secondary173 (50.0)3 (42.9)
Higher education173 (50.0)4 (57.1)
Missing955
**Marital status**

1.000Single or never married39 (11.0)0
Married or living as married258 (73.1)6 (85.7)
Divorced, separated, or widowed56 (15.9)1 (14.3)
Missing255
**Household composition**

0.356Living with others283 (80.9)7 (100.0)
Living alone67 (19.1)0
Missing555
**Cancer type**

0.809Breast154 (43.4)26 (41.9)
Genitourinary56 (15.8)12 (19.4)
Gynecological51 (14.4)5 (8.1)
Melanoma46 (13.0)9 (14.5)
Head and neck21 (5.9)4 (6.5)
Soft tissue and bone15 (4.2)3 (4.8)
Other12 (3.4)3 (4.8)
Missing00
**Therapy intention**

0.246Palliative279 (78.6)44 (71.0)
Curative76 (21.4)18 (29.0)
Missing00
**Estimated life expectancy**

0.763>1 year273 (76.9)46 (74.2)
≤1 year82 (23.1)16 (25.8)
Missing00
**Therapy change**

0.331Yes41 (11.5)4 (6.5)
No314 (88.5)58 (93.5)
Missing00
Data are absolute frequency (%) or median (IQR). The participant cohort includes individuals who participated in at least one survey. The χ^2^ test for categorical variables (or the Fisher exact test when expected counts were less than five) and the two-sample t test for continuous variables were used for the comparisons between groups.

Figure 4A,B shows the longitudinal course of mental health scores (treated as continuous variables) among participants; corresponding numerical values for all datapoints are shown in Appendix A. Changes from baseline in mental health scores are presented in Table 2; the MID was not reached for any of these scores at T1, T2, or T3.

### 3.3. Quality-of-Life Scores

Figure 4C,D shows the longitudinal course of QOL scores among participants; corresponding numerical values for all datapoints are shown in Appendix A. Changes from baseline in QOL scores are presented in Table 2; the MID was not reached for any of these scores at T1, T2, or T3.

### 3.4. Factors Associated

#### 3.4.1. With Mental Health Scores

Appendix A shows the results of the bivariate analyses for mental health scores. Adjusted analyses using multivariable linear mixed regression modeling showed that the covariates independently associated with worse mental health scores were (Table 3): for depressive symptoms, living alone (*p* = 0.035); and for anxiety, male gender (*p* = 0.001), living alone (*p* = 0.020), and an estimated life expectancy of ≤1 year (*p* = 0.004). No statistically significant independent associations were found for COVID-19 peritraumatic distress and stress.

#### 3.4.2. With Quality-Of-Life Scores

Appendix A shows the results of the bivariate analyses for QOL scores. Adjusted analyses using multivariable linear mixed regression modeling showed that the covariates independently associated with worse QOL were (Table 4): for physical health, marital status (*p* = 0.015; no significant pairwise comparisons), cancer type (*p* = 0.049; no significant pairwise comparisons), and an estimated life expectancy of ≤1 year (*p* = 0.001); for psychological health and younger age (*p* = 0.018); for social relationships, gender (*p* < 0.001), marital status (*p* = 0.012; no significant pairwise comparisons), living alone (*p* = 0.028), palliative intention (*p* = 0.015), and an estimated life expectancy of ≤1 year (*p* = 0.049); for environment, fundamental or secondary education (*p* = 0.002), marital status (*p* = 0.006; no significant pairwise comparisons), and cancer type (*p* = 0.027; no significant pairwise comparisons); and for overall QOL and general health, cancer type (*p* = 0.021; no significant pairwise comparisons), and an estimated life expectancy of ≤1 year (*p* < 0.001).
Figure 4Mental health and quality-of-life scores by index week of 2020. (**A**) Peritraumatic distress, (**B**) depression, anxiety, and stress, (**C**,**D**) and quality of life scores plotted against the week index of 2020. T0 was situated in weeks 14 and 15, T1 in week 20, T2 in week 26, and T3 in week 38. Score data are means; error bars show the 95% confidence intervals. Corresponding numerical values for all datapoints are shown in Appendix A. (**E**) Total number of lab-confirmed hospitalized COVID-19 patients in Belgium; vertical dotted reference lines correspond to survey periods T0 to T3, respectively; grey zone indicates the period of national lockdown (week 12 to 20). QOL: quality of life.
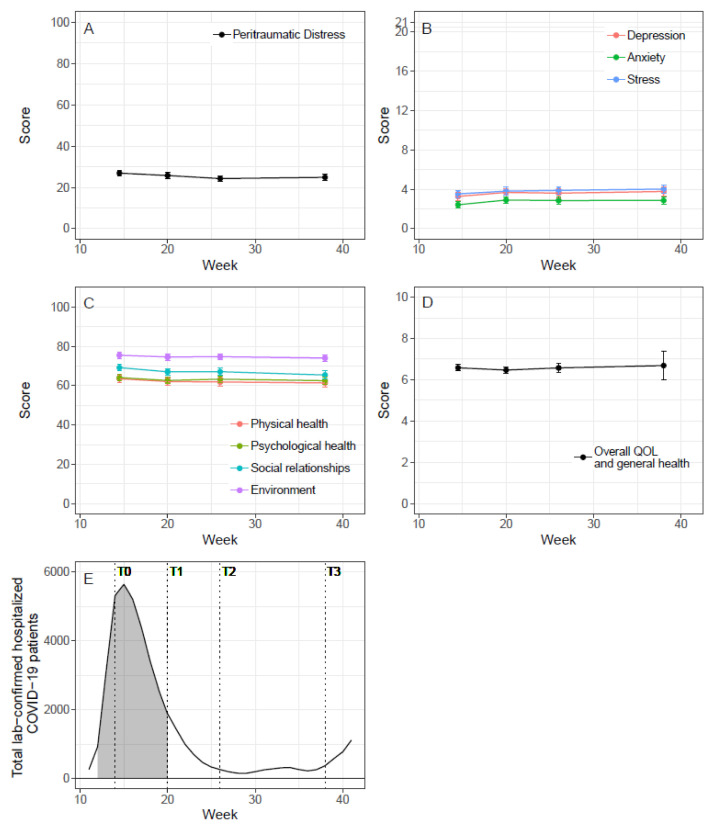

cancers-14-01093-t002_Table 2Table 2Changes from baseline in mental health and quality-of-life scores.
Survey Period
VariableT1-T0T2-T0T3-T0MID**Mental health**



COVID-19 peritraumatic distress (0–100)−1.16 (−2.22 to −0.11; ***p* = 0.028**)−2.65 (−3.73 to −1.57; ***p* < 0.001**)−2.04 (−3.26 to −0.83; ***p* < 0.001**)6.54Depression (0–21)0.41 (0.11 to 0.71; ***p* = 0.007**)0.33 (−0.02 to 0.69; *p* = 0.059)0.51 (0.14 to 0.88; ***p* = 0.006**)1.85Anxiety (0–21)0.48 (0.24 to 0.72; ***p* < 0.001**)0.43 (0.13 to 0.73; ***p* = 0.004**)0.44 (0.08 to 0.80; ***p* = 0.015**)1.44Stress (0–21)0.28 (0.04 to 0.60; *p* = 0.081)0.36 (0.02 to 0.70; ***p* = 0.037**)0.49 (0.07 to 0.91; ***p* = 0.019**)1.82**QOL**



Physical health (0–100)1.40 (−2.86 to 0.05; *p* = 0.054)−1.64 (−3.25 to −0.04; ***p* = 0.041**)−2.12 (−3.91 to −0.32; ***p* = 0.019**)9.42Psychological health (0–100)−1.48 (−2.66 to −0.29; ***p* = 0.013**)−0.80 (−2.06 to 0.45; *p* = 0.201)−1.58 (−3.10 to −0.07; ***p* = 0.037**)7.44Social relationships (0–100)−2.13 (−3.54 to −0.72; ***p* = 0.003**)−2.08 (−3.54 to −0.61; ***p* = 0.005**)−3.75 (−5.47 to −2.03; ***p* < 0.001**)8.04Environment (0–100)−0.85 (−1.95 to 0.25; *p* = 0.122)−0.72 (−1.94 to 0.51; *p* = 0.240)−1.35 (−2.69 to 0.002; ***p* = 0.046**)6.96Overall QOL and general health (0–10)−0.11 (−0.25 to 0.03; *p* = 0.114)−0.004 (−0.24 to 0.23; *p* = 0.970)0.11 (−0.59 to 0.81; *p* = 0.753)0.80Data are mean absolute difference (with 95% CI; *p* Value) from baseline (T0). The minimal important differences are displayed in the far most right column. MID: minimal important difference; QOL: quality of life.
cancers-14-01093-t003_Table 3Table 3Factors associated with mental health identified by multivariable mixed linear regression analysis.
Mental Health
CPDIDASS-21VariablePeritraumatic Distress*p* ValueDepression*p* ValueAnxiety*p* ValueStress*p* Value**Age**−0.06 (−0.17 to 0.05)0.426−0.02 (−0.05 to 0.01)0.584−0.003 (−0.03 to 0.02)0.239−0.02 (−0.05 to 0.009)0.291**Gender**
0.085
0.728
0.005
0.770MaleReference
Reference
Reference
Reference
Female0.67 (−3.66 to 4.99)
0.51 (−0.76 to 1.79)
−0.97 (−1.93 to −0.009)
−0.28 (−1.45 to 0.90)
**Educational level**
0.073
0.757
0.225
0.465Fundamental or secondaryReference
Reference
Reference
Reference
Higher education−2.38 (−5.06 to 0.29)
−0.19 (−0.97 to 0.60)
−0.40 (−0.99 to 0.19)
−0.25 (−098 to 0.47)
**Marital status**
0.382
0.685
0.602
0.663Single or never marriedReference
Reference
Reference
Reference
Married or living as married3.58 (−1.89 to 9.04)
0.87 (−0.73 to 2.47)
0.74 (−0.46 to 1.94)
1.25 (−0.23 to 2.73)
Divorced, separated, or widowed3.69 (−1.59 to 8.97)
0.32 (−1.23 to 1.87)
0.27 (−0.89 to 1.43)
0.46 (−0.97 to 1.89)
**Household composition**
0.428
0.035
0.020
0.140Living aloneReference
Reference
Reference
Reference
Living with others−2.25 (−7.43 to 2.92)
−1.62 (−3.14 to −0.09)
−1.39 (−2.53 to −0.24)
−1.05 (−2.46 to 0.36)
**Cancer type**
0.567
0.377
0.344
0.796BreastReference
Reference
Reference
Reference
Genitourinary−3.24 (−8.69 to 2.21)
0.77 (−0.84 to 2.37)
−0.20 (−1.40 to 1.00)
−0.43 (−191 to 1.05)
Gynecological−0.56 (−4.51 to 3.39)
−0.20 (−1.35 to 0.96)
−0.40 (−1.27 to 0.46)
0.29 (−0.77 to 1.36)
Melanoma−4.89 (−9.59 to −0.19)
−0.65 (−2.04 to 0.74)
−0.92 (−1.96 to 0.13)
−0.61 (−1.89 to 0.68)
Head and neck−1.94 (−8.30 to 4.42)
0.25 (−1.66 to 2.17)
0.13 (−1.31 to 1.57)
−1.06 (−2.83 to 0.72)
Soft tissue and bone1.86 (−5.24 to 8.96)
1.64 (−0.43 to 3.71)
0.39 (−1.15 to 1.94)
0.85 (−1.06 to 2.77)
Other−2.60 (−9.98 to 4.77)
0.55 (−1.60 to 2.70)
−0.44 (−2.04 to 1.16)
−0.76 (−2.74 to 1.22)
**Therapy intention**
0.939
0.843
0.487
0.687CurativeReference
Reference
Reference
Reference
Palliative−0.13 (−3.58 to 3.31)
0.10 (−0.91 to 1.11)
−0.27 (−1.02 to 0.49)
0.19 (−0.74 to 1.13)
**Estimated life expectancy**
0.052
0.120
0.004
0.328≤1 yearReference
Reference
Reference
Reference
>1 year−2.79 (−6.08 to 0.51)
−0.60 (−1.57 to 0.37)
−0.95 (−1.68 to −0.22)
−0.30 (−1.20 to 0.59)
**Therapy change**
0.734
0.619
0.324
0.267NoReference

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

0.835
0.965
**<0.001**
0.290
0.460MaleReference
Reference
Reference
Reference
Reference
Female−1.40 (−7.41 to 4.60)
−4.37 (−9.29 to 0.56)
4.71 (−0.23 to 9.66)
−1.77 (−6.22 to 2.68)
−0.08 (−0.58 to 0.41)
**Educational level**
0.534
0.696
0.681
**0.002**
0.766Fundamental or secondaryReference
Reference
Reference
Reference
Reference
Higher education1.20 (−2.51 to 4.91)
0.69 (−2.36 to 3.74)
0.92 (−2.14 to 3.98)
4.30 (1.55 to 7.06)
−0.05 (−0.35 to 0.26)
**Marital status**
**0.015***^i^*
0.268
**0.012***^ii^*
**0.006***^iii^*
0.124Single or never marriedReference
Reference
Reference
Reference
Reference
Married or living as married−2.34 (−9.92 to 5.23)
−3.61 (−9.84 to 2.62)
−1.90 (−8.15 to 4.34)
−1.89 (7.52 to 3.73)
−0.26 (−0.8 to 0.36)
Divorced, separated, or widowed−8.17 (−15.49 to −0.84)
−4.08 (−10.10 to 1.94)
−4.34 (−10.38 to 1.71)
−6.89 (−12.32 to −1.46)
−0.57 (−1.17 to 0.03)
**Household composition**
0.473
0.143
**0.028**
0.546
0.618Living aloneReference
Reference
Reference
Reference
Reference
Living with others2.75 (−4.43 to 9.93)
4.71 (−1.18 to 10.60)
6.22 (0.30 to 12.14)
1.74 (−3.58 to 7.07)
0.15 (−0.44 to 0.74)
**Cancer type**
**0.049**^*iv*^
0.172
0.075
**0.027**^*v*^
**0.003**^*vi*^BreastReference
Reference
Reference
Reference
Reference
Genitourinary−2.63 (−10.18 to 4.92)
−7.61 (−13.81 to −1.41)
−4.92 (−11.14 to 1.29)
−5.53 (−11.13 to 0.07)
−0.28 (−0.90 to 0.34)
Gynecological2.90 (−2.57 to 8.37)
−0.19 (−4.68 to 4.30)
−0.02 (−4.53 to 4.49)
−0.83 (−4.89 to 3.23)
−0.12 (−0.57 to 0.33)
Melanoma2.70 (−3.83 to 9.22)
−2.21 (−7.56 to 3.13)
−2.42 (−7.79 to 2.95)
−1.44 (−6.28 to 3.39)
0.34 (−0.20 to 0.87)
Head and neck0.10 (−8.77 to 8.98)
−4.03 (−11.29 to 3.22)
−0.53 (−7.87 to 6.80)
−5.84 (−12.41 to 0.72)
−0.65 (−1.38 to 0.08)
Soft tissue and bone−12.91 (−22.74 to −3.09)
−7.96 (−16.04 to 0.11)
−12.84 (−20.93 to −4.75)
−8.57 (−15.86 to −1.28)
−1.05 (−1.85 to −0.24)
Other−1.93 (−12.13 to 8.28)
−3.18 (−11.56 to 5.20)
5.48 (−2.95 to 13.91)
0.05 (−7.53 to 7.64)
−0.45 (−1.28 to 0.38)
**Therapy intention**
0.981
0.185
**0.015**
0.764
0.688CurativeReference
Reference
Reference
Reference
Reference
Palliative−0.06 (−4.82 to 4.71)
2.65 (−1.27 to 6.56)
−4.90 (−8.82 to −0.97)
0.54 (−2.99 to 4.08)
−0.08 (−0.47 to 0.31)
**Estimated life expectancy**
**0.001**
0.052
**0.049**
0.051
**<0.001**≤1 yearReference
Reference
Reference
Reference
Reference
>1 year6.60 (2.03 to 11.17)
3.85 (0.10 to 7.60)
1.59 (−2.18 to 5.36)
2.82 (−0.57 to 6.21)
0.73 (0.36 to 1.11)
**Therapy change**
0.141
0.357
0.138
0.348
0.398NoReference
Reference
Reference
Reference
Reference
Yes−4.29 (−10.10 to 1.53)
−1.84 (−6.60 to 2.93)
−4.15 (−8.94 to 0.65)
−1.94 (−6.25 to 2.37)
−0.21 (−0.69 to 0.27)
Data are expressed as adjusted regression coefficients (95% confidence interval [CI]). Bonferroni-corrected *post hoc* thresholds were applied after obtaining significant results: *^i^* significant differences in pairwise comparisons (threshold *p* < 0.01667): none; *^ii^* significant differences in pairwise comparisons (threshold *p* < 0.01667): none; *^iii^* significant differences in pairwise comparisons (threshold *p* < 0.01667): none; *^iv^* significant differences in pairwise comparisons (threshold *p* < 0.0024): none; *^v^* significant differences in pairwise comparisons (threshold *p* < 0.0024): none. *^vi^* significant differences in pairwise comparisons (threshold *p* < 0.0024): none.

## 4. Discussion

This longitudinal survey study examined the course of multiple mental health symptoms and QOL during the SARS-CoV-2 pandemic in a cohort of patients with cancer receiving systemic anticancer therapy (at least during T0). Contrary to our hypothesis, we found neither evidence that there was a clinically significant increase in mental health symptoms nor that there was a clinically significant decrease in QOL as the SARS-CoV-2 pandemic continued. Thus, our sample of cancer patients showed considerable resilience against mental health and QOL deterioration during the SARS-CoV-2 pandemic. 

It is critical to emphasize that our data collection started relatively early in the pandemic (T0; 1 April to 10 April 2020) (Figure 2E). Prior to the beginning of the T0 survey period, the Belgian government mandated a national lockdown, which included obligated telecommuting and the closure of borders, schools, and non-essential commercial establishments. The highest number of lab-confirmed hospitalized COVID-19 patients (5759) during the pandemic’s first wave was recorded on 6 April 2020, followed by a steady decrease. Thus, T0 captured the peak of the pandemic’s first wave, whereas T1 (11 May to 15 May 2020) was situated in the subsequent deceleration stage. By contrast, T2 (22 June to 26 June 2020) was situated in a relatively calm period of easing restrictions, with further declining hospital admissions. Finally, T3 (14 September to 18 September 2020) coincided with the second week of the pandemic’s second wave as the number of hospitalizations was increasing rapidly. Therefore, our results are important because—to our knowledge—they are among the first to longitudinally assess mental health symptoms and QOL in patients with cancer during different stages of the SARS-CoV-2 pandemic. Other strengths include a well-characterized participant cohort, rather high survey response rates (between 61.5% and 79.4%), a relatively low likelihood of nonresponse bias, and the use of a COVID-19-specific questionnaire as well as questionnaires of depressive symptoms, anxiety, stress, and several QOL domains to assess multiple dimensions of the patients’ response to the pandemic. 

An important proportion of the participants reported symptoms of COVID-19 peritraumatic distress (34.2% to 39.6%), depression (27.6% to 33.5%), anxiety (24.9% to 32.7%), and stress (11.4% to 15.7%) at any time point during the study period. Although many participants experienced mental health symptoms, it should be noted that our results provide more grounds for optimism than those reported in other studies. Although some individual studies reported high prevalence rates for depressive symptoms (71.2%), anxiety (78.0%), and stress (36.0%) among cancer patients during the pandemic [16], the pooled prevalences in two recent meta-analyses were lower yet still substantial: 37% to 43% for depressive symptoms and 38% to 53% for anxiety [17,18]. In fact, our results seem more similar to those observed among the general population during the pandemic; the pooled prevalences of depressive symptoms, anxiety, and stress in this setting were 33.7%, 31.9%, and 29.6%, respectively, in another meta-analysis [31]. Such estimates should, however, be interpreted with caution. First, the estimated prevalences of mental health symptoms vary heavily according to the population studied and methodology used; thus, it is difficult to compare our results with the estimates of other studies. Therefore, as mentioned before, a key strength of this study is that it was designed to measure dynamics, rather than a single static estimate obtained at an individual point during the SARS-CoV-2 pandemic. Second, abnormal CPDI or DASS-21 scores do not necessarily constitute psychopathology per se (such as syndromal depression or anxiety disorder), but rather represent an increased severity and/or extent of symptoms. 

When examining longitudinal patterns for DASS-21 subscales, we found particular patterns of symptom change. The slight, but statistically significant, increase in depressive symptoms and anxiety (and the nearly significant increase for stress) observed at T1 may have been due to sensitization and prolonged exposure to the pandemic. One has to bear in mind that cautious relaxations of the national lockdown measures were just being implemented at T1. By contrast, the stabilization/recovery at T2 might be explained by habituation to the COVID-19 crisis and the resumption of (non-urgent) mental health services, whereas the subsequent increase at T3 might have been the result of the second wave gathering momentum and/or the corresponding containment measures. Overall, although not clinically significant, our findings imply a prolonged deterioration in depressive symptoms, anxiety, and stress during the pandemic. This is consistent with the results of a large and high-quality probability-based sample of the adult UK general population, which reported a small rapid decline in mental health that did not begin improving until July 2020 [32]. Similarly, a recent meta-analysis also found a small increase in mental health symptoms among the general population soon after the outbreak of the pandemic that subsequently decreased and was comparable to prepandemic levels by mid-2020 [33].

Because our study was designed in response to the SARS-CoV-2 pandemic (and direct comparison was not a study objective), no contemporaneous control group was available. Therefore, we were unable to unequivocally distinguish the contribution of the events associated with the SARS-CoV-2 pandemic versus other ecological drivers (that would have occurred anyway) on changes in mental health. In this context, it has to be noted that one cross-sectional study among 306 patients with cancer actively treated with systemic therapy found that SARS-CoV-2-associated fear and anxiety were significantly lower than cancer-associated anxiety during the period of serious restrictions. However, given that it is possible that the mental health impact of and coping with COVID-19-related events need time to accrue [34,35], the mentioned study’s cross-sectional design did not permit the assessment of whether trajectories of increasing SARS-CoV-2-associated fear and anxiety emerged as the pandemic continued. Moreover, given the ubiquitous nature and vast extent of the pandemic, we are rather confident in the ecological validity of our findings, and thus, that the reported changes are likely to be truly pandemic-related. 

The CPDI, however, enabled us to directly capture the mental health impact of the pandemic. The self-report measures of COVID-19 peritraumatic distress indicated a pattern of immediate recovery, followed by a deterioration at T3. While this finding might seem to contradict the pattern of depressive symptoms, anxiety, and stress that showed an initial increase, the CPDI score actually followed the total number of COVID-19 hospitalizations well. One possible explanation for the differing results is that the CPDI might better (in contrast to the DASS-21) distinguish which mental health effects are attributable to the pandemic itself and which mainly result from containment measures. However, an alternative explanation could be that the etiology of depression, anxiety, and stress are multifactorial, whereas COVID-19 peritraumatic distress is not (by definition). Finally, it should be noted that the CPDI has not yet been validated in Dutch, although studies have been initiated. 

The WHOQOL-BREF scores generally followed similar longitudinal patterns as the DASS-21 scores, with an initial deterioration followed by subsequent stabilization and second deterioration; however, the differences were not clinically significant at any time point during the study. Our study purposedly focused on QOL broadly, rather than health-related QOL; the WHOQOL-BREF assessed contextual variables that are not generally regarded as health-related (e.g., satisfaction with transport, leisure activities, social support, and financial resources) but that are particularly relevant in the context of a pandemic. Unsurprisingly, we found that the largest deterioration occurred in the QOL domain of social relationships at T3 that, despite being statistically significant, was not clinically important. Given that the practice of physical distancing (often incorrectly referred to as ‘social’ distancing) may narrow the sense of social connection [36], we anticipated a greater deterioration in this domain. However, this finding corroborates those of previous studies in the general population that suggest that the impact on social connection and loneliness was modest [37,38]. One explanation for our finding is that the perceived impact of COVID-19, which was disproportionately large for cancer patients, has been shown to be associated with less loneliness and greater perceptions of social support [39,40]. An alternative explanation might be that a large proportion of cancer patients can count on psychological and/or social support by various professionals and others (regardless of the pandemic), including members of the community, friends, and family [41]. Indeed, it has been reported that the quality and quantity of social relationships is positively related to mental health and well-being during the pandemic [42,43]. Finally, there is anecdotal evidence that social distancing was already adhered to during the prepandemic period. Hence, the impact of the pandemic on cancer patients (especially those receiving active therapy) may might have been minimal.

We also aimed to identify protective or risk factors for those who fared better or worse, respectively, during the pandemic. Overall, there were no consistent correlates of mental health or QOL scores. Although multiple baseline characteristics were independently associated with mental health or QOL scores in the present study, their regression coefficients were clinically insignificant as the accompanying 95% CIs contained the MID. In addition, these analyses were neither powered nor corrected for multiple comparisons (except *post hoc* pairwise comparisons) and should be interpreted with caution. We acknowledge that some unmeasured characteristics of participants (e.g., personality and economic support) have also been identified as protective or risk factors for well-being during the pandemic in the general population [44,45,46]. Nevertheless, a meta-analysis on prepandemic prevalences of depression, anxiety, and adjustment disorders in patients with cancer also found no predictors of prevalence, including age, cancer type, clinical setting, illness duration, or sex [6]. 

There are several limitations to this study, some of which are due to the logistical challenges of designing and conducting research during a pandemic. The most important limitations are that the study took place at only one tertiary care hospital and did not include a control group, nor did it contain pre-COVID-19 baseline data against which to measure change. Accordingly, the design did not enable us to investigate differences with other groups (e.g., healthy individuals) or changes from the prepandemic period. Second, patients with gastrointestinal and thoracic cancers were underrepresented in our sample, thus explaining the small proportion of male participants. Third, surveys were carried out during predefined time periods; our data could thus be influenced by highly situational factors (e.g., growth versus decay phase of the pandemic, tightening versus relaxation of the containment measures [35], and exposure to [sensationalized] media coverage [47]). Fourth, there could also be geographic factors: the hospital was located in Belgium, which suffered some of the highest per capita COVID-19 death rates in the world. This, in combination with the stringent and disruptive socio-economic containment measures in Belgium, could limit the generalization of our findings to other (sub-)regions [48,49].

## 5. Conclusions

In summary, there was a considerable burden of mental health symptoms during the SARS-CoV-2 pandemic in our cohort of cancer patients. However, these cancer patients were arguably faring better than expected. In particular, the present study did not find clinically meaningful patient-reported outcome differences during the study period, with remarkably little change in between the first and second wave. Nevertheless, clinicians should remain vigilant for mood complications. Moreover, during the continuation of the pandemic in the future (or similar events), the oncological community should rely on empirical data rather than theoretical assumptions to inform interventions and policies that balance physical health on one side and mental health and quality of life on the other.

## Data Availability

The datasets generated and/or analyzed during the current study are available from the corresponding author on reasonable request.

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
