# Peer review of "Mental Health and Quality of Life among Patients with Cancer during the SARS-CoV-2 Pandemic: Results from the Longitudinal ONCOVID Survey Study"

_cancers, 2022, doi:10.3390/cancers14041093_

Round 1

Reviewer 1 Report

  1. Introduction is lengthy - by now we ought to know what SARS-CoV-2 is. 
  2. T0 is april 2020 and unfortunately this means that a diference is stress with or without SARS-CoV2 can not be filtered out. Overall Stress (still 34-39%) does not seem to be progressing over time (several surges of Covid) and QoL is primarily influenced by tumor-related parameters. 
  3. in the discussion I would suggest to point out that the observations made are particular to a National setting and therefore somewhat difficult to compare since health insurance or care management can differ according to region or country

Author Response

  1. General review

We would like to thank Reviewer 1 for his/her/them time. Below, we present our answers to the comments in a point-by-point fashion.

  1. English language and style are fine/minor spell check required

The manuscript was revised by a native speaker; grammatical and/or punctuation errors were corrected.

  1. Introduction is lengthy - by now we ought to know what SARS-CoV-2 is.

We fully agree with Reviewer 1 that many researchers are familiar with SARS-CoV-2; however, research manuscripts should withstand the test of time. Therefore, we question whether (perhaps young and new) researchers still ought to know SARS-CoV-2 in ten (or more) years? As the generic introduction on SARS-CoV-2 only encompasses eight lines, we believe it is of sufficient importance to include it in the manuscript. The length of the entire introduction is less than one page and reports all elements as required by the STROBE statement.

  1. T0 is april 2020 and unfortunately this means that a diference is stress with or without SARS-CoV2 can not be filtered out. Overall Stress (still 34-39%) does not seem to be progressing over time (several surges of Covid) and QoL is primarily influenced by tumor-related parameters.
  • Thank you for this observation. We agree with Reviewer 1 that this is an important aspect of the discussion.
    • Unfortunately, direct comparison to prepandemic levels was not possible, as shown on lines 407-420. In addition, we also referred to the importance of tumor-related parameters (“cancer-associated anxiety”) in the same section.

“Because our study has been designed in response to the SARS-CoV-2 pandemic (and direct comparison was not a study objective), no contemporaneous control group was available. Therefore, we were unable to unequivocally distinguish the contribution of the events associated with the SARS-CoV-2 pandemic versus other ecological drivers (that would have occurred anyway) on changes in mental health. In this context, it has to be noted that one cross-sectional study among 306 patients with cancer actively treated with systemic therapy found that SARS-CoV-2–associated fear and anxiety were significantly lower than cancer-associated anxiety during the period of serious restrictions. However, given that it is possible that the mental health impact of and coping with COVID-19–related events need time to accrue [34,35], the mentioned study’s cross-sectional design did not permit assessing whether trajectories of increasing SARS-CoV-2–associated fear and anxiety emerged as the pandemic continued. Moreover, given the ubiquitous nature and vast extent of the pandemic, we are rather confident in the ecological validity of our findings, so that the reported changes are likely to be truly pandemic-related.”

  • Furthermore, we also mentioned this in the limations section of our discussion (lines 468-471).

“The most important limitations are that the study took place at only one tertiary care hospital and did not include a control group, nor did it contain pre-COVID-19 baseline data against which to measure change. Accordingly, the design did not enable us to investigate differences with other groups (e.g., healthy individuals) or changes from the prepandemic period.”

  • Regardless of all the above, the CPDI questionnaire in our study enabled us to directly capture the mental health impact of the pandemic as it was designed specifically for this purpose (as mentioned on lines 421-432).

“The CPDI, however, enabled us to directly capture the mental health impact of the pandemic. The self-report measures of COVID-19 peritraumatic distress indicated a pattern of immediate recovery, followed by a deterioration at T3. While this finding might seem to contradict the pattern of depressive symptoms, anxiety, and stress that shows an initial increase, the CPDI score actually followed the total number of COVID-19 hospitalizations well. One possible explanation for the differing results is that the CPDI might better (in contrast to the DASS-21) distinguish which mental health effects are attributable to the pandemic itself and which mainly result from containment measures. However, an alternative explanation could be that the etiology of depression, anxiety, and stress are multifactorial, whereas COVID-19 peritraumatic distress is not (by definition). Finally, it should be noted that the CPDI has not yet been validated in Dutch, although studies have been initiated. “

  • Finally, we agree that overall stress (and the other mental health outcomes: depressive symptoms, anxiety, and covid-19 peritraumatic distress) does not seem to change throughout the study. This is a actually a major finding of our study as it captures the longitudinal evolution of mental health and quality-of life outcomes (this manuscript is [among] the first to do so). Therefore, we have stressed this on several occasions throughout the manuscript (lines 348-352, 364-366, and 483-484, respectively [but on many more occasions])

“Contrary to our hypothesis, we did neither find evidence that there was a clinically significant increase in mental health symptoms nor a clinically significant decrease in QOL as the SARS-CoV-2 pandemic continued. Thus, our sample of cancer patients showed considerable resilience against mental health and QOL deterioration during the SARS-CoV-2 pandemic.”

“Therefore, our results are important because—to our knowledge—they are among the first to longitudinally assess mental health symptoms and QOL in patients with cancer during different stages of the SARS-CoV-2 pandemic.”

“In particular, the present study did not find clinically meaningful patient-reported outcome differences during the study period, with remarkably little change in between the first and second wave.”

  1. In the discussion I would suggest to point out that the observations made are particular to a National setting and therefore somewhat difficult to compare since health insurance or care management can differ according to region or country.

We thank Reviewer 1 for this excellent suggestion. We have made some adaptations in the limations section of the discussion to elaborate on this drawback (lines 478-481).

“Fourth, there could also be geographic factors: the hospital was located in Belgium, which suffered among the highest per capita COVID-19 death rates in the world. This, in combination with the stringent and disruptive socio-economic containment measures in Belgium, could limit the generalization of our findings to other (sub-)regions [48,49].”

Reviewer 2 Report

Thank you for the opportunity to review. The manuscript is well written and clearly presented. Below please see my specific comments.

  1. Section 2.2. Since this is a longitudinal study, please make it clear that , COVID infection was only check at initial enrollment and at every time point. Also, what is the rationale for excluding patients with confirmed or clinically suspected COVID infection? This study is about their mental health.
  2. Line 155, please specify equivalent grade level or degree for the 3 education levels. For instance, what is considered a higher education? Is it college or above?
  3. There is no power calculation. Please add.
  4. Lines 183. Standard deviation could change for measured at different time point. Which standard deviation was used? The one at T0, or pooled standard deviation using all measures at all points? Please clarify.
  5. QOL has limited range 0-10. May consider transform it to 100 as other scores for analysis.
  6. Line 190, “We examined descriptive statistics to ensure that data met statistical (test) assumptions”. This is not clear. Please explain what descriptive statistics were performance and to test what assumptions.
  7. Figure 1 is confusing. The number of death appear cumulative. It may be better to present the additional number of deaths at each time period since the arrow was from each segment of time. Also, with the numbers in the excluded boxes, I could not get the number of responders for 6-w, 12-w, 24-w. Suggest the exclusion boxes to include only incremental changes in numbers from the previous time point. Or, the authors could provide footnote to help readers understand those numbers.
  8. Line 203, “approached” sounded like that a person has been approached for enrolling in the study. But 8 of 452 may be moribund or died, which means they could not be approached. Consider changing to another word, like “screened” etc. These patients were screened consecutively during what time period? Were the 35 excluded after a chart review?
  9. Figure 2 and 3. Are these generated using all available data? Suggest also presenting the results using only those who filled all 4 surveys on the side.
  10. Line 250. There are 4 subscales and should not use “either” scale. Please revise the sentence.
  11. Please add MID or include the baseline mean and SD scores in Table 2 for easy comparison and assessment the magnitude (i.e. relative to SD) of the changes overall time. This information is currently reported in supplement table but should be added to Table 2. Are the numbers in Table 2 adjusted means from LMM?
  12. Line 128. Please provide additional information on the validity of CPDI questionnaire. It seems it could overlap with other mental health scales. Could the author provide some discussion about that? Why it is necessary for use all 4 scales?
  13. Lines 280-284. Some of the p-values are not consistent with Table 3. Not all significant variables are reported in this section. For instance, cancer type and estimated life expectancy are also significant for several scales.
  14. In the discussion of the prevalence of these mental health conditions with previous literature, please also provide the time periods when other studies were conducted. Please provide some insight why this study results are much lower than a previous studies of cancer patients. Are previous studies used confirmed diagnosis of these conditions?
  15. Lines 445, is their any reference for the statement that “the perceived impact of COVID 19 is disproportionately large for cancer patients”? This study’s findings do not seem to support that.  There are also anecdotal evidence that pandemic measures like social distancing and mask wearing (especially of those receiving active treatment) are already an accepted way of life for patients with cancer or those with compromised immune system. So, the impact of pandemic on cancer patients may be minimum. It may help explain what is observed in this study.

Author Response

  1. Thank you for the opportunity to review. The manuscript is well written and clearly presented.

We are grateful for this comment as it points out that although the manuscript should be revised,  its findings are of interest to the readership of Cancers. Moreover, we sincerely thank Reviewer 2 for his/her/them time; we believe the suggestions made by Reviewer 2 improved the overall quality of our manuscript. Below, we present our answers to the comments in a point-by-point fashion.

  1. English language and style are fine/minor spell check required.

The manuscript was revised by a native speaker; grammatical and/or punctuation errors were corrected.

  1. Section 2.2. Since this is a longitudinal study, please make it clear that , COVID infection was only check at initial enrollment and at every time point. Also, what is the rationale for excluding patients with confirmed or clinically suspected COVID infection? This study is about their mental health.

When designing this study, it was unsure whether coronavirus-2019 could spread through surfaces (e.g., paper). Therefore we wanted to assure that our research team could safely handle all documents. Please note that only 3 patients were excluded during the entire study. Patients were asked whether they were infected (or not) at every survey period.

  1. Line 155, please specify equivalent grade level or degree for the 3 education levels. For instance, what is considered a higher education? Is it college or above?

We agree with the reviewer that this needs some clarification as education system terminology can vary greatly across borders. The Belgian eduction system is divided in four parts: preschool (ages, 2 to 6), primary school (6 to 12), secondary eduction (12 to 18), and tertiary eduction (university or nonuniversity formats; from age 18 on, typically taking 2 to 5 years to complete). The adaptations are underlined.

“Highest educational level was defined as a categorical variable with two groups: fundamental or secondary education (preschool, primary school, and secondary eduction); and higher education (university and nonuniversity formats).”

  1. There is no power calculation. Please add.
  • Prespecified power analysis (prospective):

We gladly added the power statement from the protocol in the M&M section to adhere to this comment. Please note that our achieved sample was much larger than anticipated. We did not power our study for the secondary endpoints.

“We estimated the eligible population as 300 to 350 individuals. A response rate of 50 to 65% was targeted, which resulted in an estimated sample size of 180 patients. For the coprimary endpoints, this sample size of 180 allowed for generalizability to the study population based on a 95% Wilson score confidence interval with a half width of approximately 7.5%.”

  • Post-hoc power analysis (retrospective):
    • An insignificant result will always have low “observed/post-hoc” power. At first glance, one might conclude that the study was therefore underpowered; however, “observed/post-hoc” power is actually completely determined by the p value itself. In other words, you will always compute a low “observed/post-hoc” power on a hypothesis test with a large (non-significant) p value (Althouse A. Post Hoc Power: Not Empowering, Just Misleading. J Surg Res. 2021 Mar; 259:A3-A6; Goodman SN, Berlin JA. The use of predicted confidence intervals when planning experiments and the misuse of power when interpreting results. Ann Intern Med. 1994; 121:200-206). Therefore, we did not perform a post-hoc power analysis.
    • Nevertheless, high (non-significant) p values are not a “problem” in this study as we generally obtained highly statistically significant results. It should be noted that these statistically significant results were not clinically significant (as we elaborated in the discussion). Computing a post-hoc power would therefore not provide any additional information (even when ignoring the first bullet point).

  1. Lines 183. Standard deviation could change for measured at different time point. Which standard deviation was used? The one at T0, or pooled standard deviation using all measures at all points? Please clarify.

We agree with Reviewer 2 that this statement needed some clarification (see underlined text). We used the standard deviation of the baseline data. Given that the mental health and QOL outcomes were quite stable throughout the study, it is extremely unlikely that the SD of the pooled data would have changed the conclusions.

“For all scores, differences from baseline in mean score of at least half a standard deviation (SD) (of baseline data) were considered clinically significant (minimal important difference, MID) [29].”

  1. QOL has limited range 0-10. May consider transform it to 100 as other scores for analysis.

The QOL subscale of overall quality of life and general health is based on two questions only (with each question being scored from 1 to 5). Therefore, the subscale’s raw score ranges from 2 to 10 (consisting of only integers: 2, 3, 4, …, and 10). Transforming this raw score to /100 is not recommended by the WHO (the creator of the WHOQOL-BREF questionnaire) and, in our opinion, would somewhat distort its interpretation. Please note that the transformed scores would only concist of multiples of 10 starting from 20 (20, 30, 40, …, and 100). Hence, the raw and transformed scores would be equally discrete.

  1. Line 190, “We examined descriptive statistics to ensure that data met statistical (test) assumptions”. This is not clear. Please explain what descriptive statistics were performance and to test what assumptions.

This is a generic statement related to all statistical tests. For instance, the two-sample t test for Table 1 requires several assumptions: continuous (ordinal) data, independence, data are normally distributed or a reasonably large sample is used, and homoscedasticity. Before performing this test, we assured (for instance) age was continuous, age observations in one group are independent from age observations in the other group, age was normally distributed or the sample was sufficiently large (mean=mode=mean, skewness, kurtosis, histogram, and/or simulation to assure the central limit theorem [CLT] kicks in  age was not normally distributed, yet the sample is sufficiently large for the CLT to reliably kick in to assure the sampling distribution of the mean is normally distributed), and homoscedasticity (homogeneity of variances). It is not opportune to specify all assumptions for al hypothesis tests and/or analyses, therefore this generic sentence was added (and is generally accepted in literature). All these assumptions are typically summarized by descriptive statistics (e.g., mean, kurtosis, skewness, variance, etc.). If one or more assumptions would be violated, the results may be unreliable or even misleading.

  1. Figure 1 is confusing. The number of death appear cumulative. It may be better to present the additional number of deaths at each time period since the arrow was from each segment of time. Also, with the numbers in the excluded boxes, I could not get the number of responders for 6-w, 12-w, 24-w. Suggest the exclusion boxes to include only incremental changes in numbers from the previous time point. Or, the authors could provide footnote to help readers understand those numbers.

We have taken the comment on board and clarified the caption; we believe that Figure 1 is clearer as a result.

“All the exclusion boxes in the right margin originate from the start; the numbers within these boxes should be subtracted from 390 and/or 417 to obtain the number of responders and eligible individuals at the given survey period, respectively (e.g., at T2, 291 [=390-6-20-62-11] responders out of 411 [=417-6] eligible individuals).”

  1. Line 203, "approached" sounded like that a person has been approached for enrolling in the study. But 8 of 452 may be moribund or died, which means they could not be approached. Consider changing to another word, like "screened" etc. These patients were screened consecutively during what time period? Were the 35 excluded after a chart review?
  • We thank Reviewer 2 for the suggestion; we substituted “approached” by “examined for eligibility”.
  • The patients were examined for eligibility/screened after consent was received by the Ethical Committee (27/03/2020).
  • The 35 patients were excluded after a chart review (i.e., no eligible diagnosis, no eligible therapy, cognitive impairment, moribund or diend) and after contact (i.e., unable to understand Dutch, acute psychiatric crisis, moribund or died, and COVID-19) (not mutually exclusive).

  1. Figure 2 and 3. Are these generated using all available data? Suggest also presenting the results using only those who filled all 4 surveys on the side.

Figures 2 and 3 included all available data. Because we believe that the requested additional figures (based on only those who participated in all four survey periods) do not carry additional information compare to Figures 2 and 3, we omitted them from the manuscript. However, we gladly generated and share those additional figures as requested by Reviewer 2 (see extra document).

  1. Line 250. There are 4 subscales and should not use "either" scale. Please revise the sentence.

We acknowledge that the traditional rule holds that “either” should only be used to refer to one of two items (although sometimes considered too restrictive by some). Therefore, we substituted “either” by “any”. We thank Reviewer 2 for reminding us.

  1. Please add MID or include the baseline mean and SD scores in Table 2 for easy comparison and assessment the magnitude (i.e. relative to SD) of the changes overall time. This information is currently reported in supplement table but should be added to Table 2. Are the numbers in Table 2 adjusted means from LMM?

We thank Reviewer 2 for this suggestion. The MID was added in the far most right column for easy comparison. We did not add the SD as this equals 2*MID (explicitly stated in the M&M section). Furthermore, the SD is easily found in the supplementary material. As the caption states, the data are mean absolute difference (not adjusted).

  1. Line 128. Please provide additional information on the validity of CPDI questionnaire. It seems it could overlap with other mental health scales. Could the author provide some discussion about that? Why it is necessary for use all 4 scales?

This is a valid point Reviewer 2 raises. However, some nuances should be made:

  • The longitudinal pattern changes differently for the CPDI vs the three DASS subscales (Figure 4a vs 4b; Table 2). This was discussed in lines 442-452. In short, the CPDI follows the number of COVID-19 hospitalizations quite well, whereas the DASS subscales do not. This might suggest that the CPDI is mainly influenced by mental health consequences related to COVID-19, whereas the DASS is more heavily influenced by other factors (e.g., cancer-related factors).
  • The correlation matrix (Figure S1) shows that the CPDI is somewhat correlated to the DASS subscales but not exceedingly. It is not uncommon for questionnaires to show some degree of correlation (e.g., stress and depression subscales).
  • We used the CPDI because it was specifically designed to capture COVID-19 peritraumatic distress whereas the DASS is, in our opinion, less specific (as already mentioned in the first bullet point).

  1. Lines 280-284. Some of the p-values are not consistent with Table 3. Not all significant variables are reported in this section. For instance, cancer type and estimated life expectancy are also significant for several scales.

The p values are consistent. Perhaps Reviewer 2 accidently looked at Table 4? For instance, cancer type is not statistically significantly associated with any mental health outcome in Table 3 (p values are 0.567, 0.377, 0.344, and 0.796).

  1. In the discussion of the prevalence of these mental health conditions with previous literature, please also provide the time periods when other studies were conducted. Please provide some insight why this study results are much lower than a previous studies of cancer patients. Are previous studies used confirmed diagnosis of these conditions?

We thank Reviewer 2 for these helpful comments.

  • We added the time periods of the other studies to increase interpretability. Thank you for this suggestion.
  • Indeed, we did mention estimates from previous studies. Such estimates should, however, be interpreted with caution (cf. cross-study comparisons are always inherently biased). First, the estimated prevalences of mental health symptoms vary heavily according to the population studied and methodology used; thus, it is difficult to compare our results with estimates of other studies. Therefore, as said before, a key strength of this study is that it was designed to measure dynamics, rather than a single static estimate obtained at an individual point during the SARS-CoV-2 pandemic.  We believe this was sufficiently stressed in the manuscript (lines 403-409).
  • Questionnaires in our and previous studies do not necessarily constitute psychopathology per se (such as syndromal depression or anxiety disorder), but rather represent an increased severity and/or extent of symptoms (lines 409-410). Hence, to the best of our knowledge, confirmation of diagnosis does not play a role in interpreting previous literature.

  1. Lines 445, is their any reference for the statement that “the perceived impact of COVID 19 is disproportionately large for cancer patients”? This study’s findings do not seem to support that. There are also anecdotal evidence that pandemic measures like social distancing and mask wearing (especially of those receiving active treatment) are already an accepted way of life for patients with cancer or those with compromised immune system. So, the impact of pandemic on cancer patients may be minimum. It may help explain what is observed in this study.
  • We thank Reviewer 2 for this helpful comment. The increased perceived impact was fueled by (Belgian) media coverage (i.e., having cancer was explicitly mentioned as a risk factor for increased morbidity and/or mortality) and (Belgian) oncologists (i.e., we changed therapy, switched to teleconsultations, etc. [see introduction]).
  • In addition, we have also added the explanation of Reviewer 2 in the discussion section. “Finally, there is anecdotal evidence that social distancing was already adhered to during the prepandemic period. Hence, the impact of the pandemic on cancer patients (especially those receiving active therapy) may be minimal.”

Reviewer 3 Report

In this article, the authors perform a longitudinal survey study to evaluate self-reported mental health and quality of life in patients with cancer during the phase two waves of the COVID19-pandemic. While the premise is intriguing the article suffers from some major drawbacks mentioned below.

However, each stage of the pandemic has its peculiarities that can interfere with the interpretation of such estimates

-The word ‘peculiarities’ is a bit ambiguous. The authors should clarify.

The survey period of T0 was from April 1 to 98 April 10, 2020; T1 from May 11 to May 15, 2020; T2 from June 22 to June 26, 2020; and T3 99 from September 14 to September 18, 2020.

-The authors should clarify this timeline in a graph pointing out the waves and the timepoints.

Eligible participants included individuals who were aged 18 years or older

-Was there a cut-off to this age?

-Also are these patients admitted to an in-patient facility? If yes, were they allowed visitors? I am asking because these factors can affect mental health and influence the answers in the survey.

The scores were interpreted as follows: depressive symptoms, normal (0-4), mild, (5-6), moderate (7-10), severe (11-13), and extremely severe (14-21); anxiety, normal (0-3), mild (4-5), moderate (6-7), severe (8-9), and extremely severe (10-21); and stress, normal (0-7), mild (8-9), moderate (10-12), severe (13-16), and extremely severe (17-21). These categories were based on values established in the literature [24].

-I wonder if these categories and scoring systems and values are still 1:1 translatable during pandemic-induced mental issues/ depression? The pandemic has brought in unprecedented hurdles this people of this generation have not encountered before for eg being cut off from in-person social interactions for an extended period of time.

The household composition was defined as a categorical variable with two groups: living with others and living alone.

-Were children considered as a specific category? I can assume that parents with young kids will be in greater emotional distress in some cases.

At T0, the following proportions of participants experienced symptoms of COVID-19 peritraumatic distress, depression, anxiety, and stress: 39.7%, (95% CI, 34.7 to 44.9%), 

27.6% (95% CI, 23.1 to 32.7%), 24.9% (95% CI, 20.6 to 30.0%), and 11.4% (95% CI, 8.4 to 

15.3%), respectively

-Where is this data shown?

At T0, a total of 10.9% (95% CI, 8.0 to 14.7%) of the participants experienced at least severe symptoms of either mental health score, whereas 1.5% (95% CI, 0.7 to 3.5%) experienced at least severe symptoms of all four mental health scores; at T1, 12.9% (95% CI, 9.7 to 17.0%) and 1.6% (95% CI, 0.7 to 3.6%), respectively; at T2, 14.4% (95% CI, 10.8 to 19.0%) and 0.7% (95% CI, 0.2 to 2.5%), respectively; and at T3, 12.0% (95% CI, 8.5 to 16.8%) and 1.2% (95% CI, 0.4 to 3.6%), respectively.

-There are no data callouts for this result and impossible to find.

Figure 4C-D shows the longitudinal course of QOL scores among participants; corresponding numerical values for all data points are shown in Table S3. Changes from baseline in QOL scores are presented in Table 2; the MID was not reached for any of these 263 scores at T1, T2, or T3.

-This sounds like a figure legend rather than mentioning the results.

-Figure 4 labels and captions are illegible.

-The factors from the table that are significantly correlated should be presented separately so that it is easier to follow.

However, these cancer patients 480 were arguably faring better than expected.

- Was there a control or standard that this survey data was compared with? For eg. people who did not have cancer or going through cancer therapy. How do the cancer patients fare compared to this population?

Author Response

  1. Moderate English changes required.

The manuscript was revised by a native speaker; grammatical and/or punctuation errors were corrected.

  1. In this article, the authors perform a longitudinal survey study to evaluate self-reported mental health and quality of life in patients with cancer during the phase two waves of the COVID19-pandemic. While the premise is intriguing the article suffers from some major drawbacks mentioned below.

We are grateful for this comment as it points out that although the manuscript should be revised,  its findings are of interest to the readership of Cancers. Moreover, we sincerely thank Reviewer 3 for his/her/them time; we believe the suggestions made by Reviewer 3 improved the overall quality of our manuscript. Below, we present our answers to the comments in a point-by-point fashion.

  1. However, each stage of the pandemic has its peculiarities that can interfere with the interpretation of such estimates. The word ‘peculiarities’ is a bit ambiguous. The authors should clarify.

We agree with this comment made by Reviewer 3. We believe we have clarified this now.

“However, each stage of the pandemic has its own characteristic properties that can interfere with the interpretation of such estimates (e.g., number of infections, critical care bed capacity, media coverage, etc.).”

  1. The survey period of T0 was from April 1 to 98 April 10, 2020; T1 from May 11 to May 15, 2020; T2 from June 22 to June 26, 2020; and T3 99 from September 14 to September 18, 2020. The authors should clarify this timeline in a graph pointing out the waves and the timepoints.

This was already included in Figure 4d via the vertical dotted reference lines. However, to increase clarity as requested by Reviewer 3, we (i) explicitly added labels to Figure 4d and (ii) added a new supplemental figure.

  1. Eligible participants included individuals who were aged 18 years or older
  • Was there a cut-off to this age?

There was no upper age limit. This was added on line 116-117.

  • Also are these patients admitted to an in-patient facility? If yes, were they allowed visitors? I am asking because these factors can affect mental health and influence the answers in the survey.

Unfortunately, we cannot provide an exact number to answer the first question; however, the large majority (>90%) of patients was treated in an outpatient setting (day hospital or oral therapy at home with regular consultations for toxicity checks). If the patient received therapy in the day hospital, visitors were allowed.

If the patient received oral therapy (32.2% [146/452] of the patients that were examined for eligibility), one did not have to visit the day hospital. In the latter case, teleconsultations were our preferred way of communication (especially during the first wave).

  1. The scores were interpreted as follows: depressive symptoms, normal (0-4), mild, (5-6), moderate (7-10), severe (11-13), and extremely severe (14-21); anxiety, normal (0-3), mild (4-5), moderate (6-7), severe (8-9), and extremely severe (10-21); and stress, normal (0-7), mild (8-9), moderate (10-12), severe (13-16), and extremely severe (17-21). These categories were based on values established in the literature [24]. I wonder if these categories and scoring systems and values are still 1:1 translatable during pandemic-induced mental issues/ depression? The pandemic has brought in unprecedented hurdles this people of this generation have not encountered before for eg being cut off from in-person social interactions for an extended period of time.

We believe Reviewer 3 raises an interesting point here. However, we have to work with what we have. After all, every clinical trial that was conducted during the pandemic (and that included QOL as an outcome measure) used “prepandemic” questionnaires without adjusting for the pandemic. To the best of our knowledge, we have not encountered a clinical trial adapting for this concern. Please note that we also included the CPDI which was designed during the pandemic to specifically measure COVID-19—related peritraumatic distress, and thus, provides an additional layer of information.

  1. The household composition was defined as a categorical variable with two groups: living with others and living alone. Were children considered as a specific category? I can assume that parents with young kids will be in greater emotional distress in some cases.

We made no difference whether participants lived together with adults or children. Although this is an interesting suggestion, we cannot retrospectively create an additional level of household composition and change our data. Furthermore, the addition of one  extra level would imply refitting the entire model.

  1. At T0, the following proportions of participants experienced symptoms of COVID-19 peritraumatic distress, depression, anxiety, and stress: 39.7%, (95% CI, 34.7 to 44.9%), 27.6% (95% CI, 23.1 to 32.7%), 24.9% (95% CI, 20.6 to 30.0%), and 11.4% (95% CI, 8.4 to 15.3%), respectively. Where is this data shown?

The point estimates can be deduced from Figures 2 (for CPDI) and 3 (for depression, anxiety, and stress). For instance, for the CPDI, the “moderate” and “severe” stacks at T0 equal 39.7%. The interval estimates are not shown on these figures as this would make the Figures too complex. For the same reason, the numbers were not added. As mentioned before, we want the figures and text to be complementary rather than repetitive.

  1. At T0, a total of 10.9% (95% CI, 8.0 to 14.7%) of the participants experienced at least severe symptoms of either mental health score, whereas 1.5% (95% CI, 0.7 to 3.5%) experienced at least severe symptoms of all four mental health scores; at T1, 12.9% (95% CI, 9.7 to 17.0%) and 1.6% (95% CI, 0.7 to 3.6%), respectively; at T2, 14.4% (95% CI, 10.8 to 19.0%) and 0.7% (95% CI, 0.2 to 2.5%), respectively; and at T3, 12.0% (95% CI, 8.5 to 16.8%) and 1.2% (95% CI, 0.4 to 3.6%), respectively. There are no data callouts for this result and impossible to find.

These data are not depicted in a figure. We solely report these data in the text. As figures and text should be complementary, we did not make an additional figure. It should be noted that such a figure would be rather complex. Furthermore, we already have a lot of tables and figures.

  1. Figure 4C-D shows the longitudinal course of QOL scores among participants; corresponding numerical values for all data points are shown in Table S3. Changes from baseline in QOL scores are presented in Table 2; the MID was not reached for any of these 263 scores at T1, T2, or T3.
  • This sounds like a figure legend rather than mentioning the results.
  • Figure 4 labels and captions are illegible.

We thank Reviewer 3 for pointing this out. We have increased the font size.

  • The factors from the table that are significantly correlated should be presented separately so that it is easier to follow.

All significant covariates (of the multivariate analysis) have been presented separately in the text: only significant results were written out. If the reader wants to see all, including those that are nonsignificant, results of the multivariate analysis, we refer to the table in the main document. The univariate analysis is provided in supplementary tables.

  1. However, these cancer patients 480 were arguably faring better than expected. Was there a control or standard that this survey data was compared with? For eg. people who did not have cancer or going through cancer therapy. How do the cancer patients fare compared to this population?

There was no contemporenous control group included in the study as stated in the discussion section (lines 429 and 490). We did several cross-study comparisons: i) lines 396-400, vs cancer patients during the pandemic and ii)  lines 400-403, vs general population during the pandemic. However, it should be noted that such comparisons are inherently biased (lines 403-407). We acknowledge that we therefore cannot make strong statements regarding comparisons with people who did not have cancer/not going through cancer therapy (cf. see limations section in the discussion). Nevertheless, a key strength of our study was that we studied the longitudinal patterns of both mental health and quality-of-life outcomes of one sample during different phases of the pandemic. Finally, we want to point out that “faring better than expected” is relative to our hypothesis which we formulated on lines 93-94. We link the results of the study to our hypothesis in the first section of the discussion (lines 369-370) and the discussion, per STROBE reporting guidelines.

Round 2

Reviewer 3 Report

Dear authors,

Thank you for this thoughtful rebuttal and edits. I agree with the constraints that are usually associated with a longitudinal survey study. One thought would be to add that section of your rebuttal in the discussion of the manuscript for clarity but it is not mandatory. Nevertheless, I applaud your work and presenting this interesting outcome. This article is ready for acceptance.